# Filopodial dynamics and growth cone stabilization in *Drosophila* visual circuit development

**Mehmet Neset Özel**[1,2,3]**, Marion Langen**[1†]**, Bassem A Hassan**[4,5]**,
P Robin Hiesinger**[1,2,3*]

[1]Department of Physiology, University of Texas Southwestern Medical Center, Dallas, United States; [2]Division of Neurobiology, Institute for Biology, Freie Universität Berlin, Berlin, Germany; [3]NeuroCure Cluster of Excellence, Charite Universitätsmedizin Berlin, Berlin, Germany; [4]Center for the Biology of Disease, Vlaams Instituut voor Biotechnologie, Leuven, Belgium; [5]Center for Human Genetics, University of Leuven School of Medicine, Leuven, Belgium

**\*For correspondence:** robin.
hiesinger@fu-berlin.de

**Present address:** [†]Department of Pharmaceutical Chemistry, University of California, San Francisco, San Francisco, United States

**Competing interests:** The authors declare that no competing interests exist.

**Abstract** Filopodial dynamics are thought to control growth cone guidance, but the types and roles of growth cone dynamics underlying neural circuit assembly in a living brain are largely unknown. To address this issue, we have developed long-term, continuous, fast and high-resolution imaging of growth cone dynamics from axon growth to synapse formation in cultured Drosophila brains. Using R7 photoreceptor neurons as a model we show that >90% of the growth cone filopodia exhibit fast, stochastic dynamics that persist despite ongoing stepwise layer formation. Correspondingly, R7 growth cones stabilize early and change their final position by passive dislocation. N-Cadherin controls both fast filopodial dynamics and growth cone stabilization. Surprisingly, loss of N-Cadherin causes no primary targeting defects, but destabilizes R7 growth cones to jump between correct and incorrect layers. Hence, growth cone dynamics can influence wiring specificity without a direct role in target recognition and implement simple rules during circuit assembly.

## Introduction

Live dynamics data in intact nervous systems are critical to understand developmental processes and mutant phenotypes during the establishment of synaptic connectivity. However, most analyses of molecular perturbation experiments are based on fixed tissue and most live data are obtained in cell culture. Dynamics measurements have been difficult to obtain in intact developing brains at the resolution of growth cone filopodia, especially over long developmental time periods, in any organism (*Mason and Erskine, 2000*; *Langen, et al., 2015*).

Growth cone filopodia have been shown to follow guidance cue gradients (*Gallo and Letourneau, 2004*; *Zheng, et al., 1996*) and provide physical support for growth cone migration (*Heidemann, 1990*; *Chan and Odde, 2008*). They have also been associated with dendritic spine formation (*Sekino et al., 2007*). However, filopodia may exhibit very different and changing roles during the lifetime of a growth cone (*Mason and Erskine, 2000*; *Kolodkin and Tessier-Lavigne, 2011*; *Mason and Wang, 1997*). The types and roles of filopodial dynamics that control specific growth cone behaviors during neural circuit assembly in developing brains are largely unknown.

Amongst genetic model organisms with a complex brain, *Drosophila* provides a unique combination of small size, rapid development and the ability to culture developing eye-brain complexes

**eLife digest** Genes encode complicated developmental processes, but it is clear that genetic information cannot encode each and every individual connection that forms between the nerve cells in a brain. Instead, the individual cells and nerve endings must make decisions during brain development. Up until now, few examples were known for how these nerve endings move and choose their paths and partners in a living, developing brain.

The fruit fly *Drosophila* provides a useful model to explore the 'wiring' of nerve cells in the brain, partly because a fruit fly's brain develops within a few days. However, most previous studies have relied on identifying mutant flies with disrupted brain wiring and studying them using still images. Now, Özel et al. have developed a new imaging method that has enough resolution and speed over sufficiently long periods to track the growing nerve endings in a developing fly brain. The method was applied to a model nerve cell in the fly's visual system. This revealed that most of this nerve's dynamic changes are short-lived and random, and appear to help to stabilize the developing nerve ending, rather than guide it to a target. Özel et al. also found that a protein called N-Cadherin, previously thought to be required for the targeting of developing nerve endings, actually plays a role in their stabilization.

These findings uncover the roles of changes in nerve endings during long-term brain development; this was previously largely unknown for any organism. The next stage in this research will involve further analyses of both wild type and mutant flies to try and work out general principles about how the brain develops via the decoding of genetic information.

(*Gibbs and Truman, 1998*; *Ayaz, et al., 2008*). The fly visual system provides a well-studied model for axon outgrowth, targeting, layer formation, and quantitative synapse formation (*Hadjieconomou et al., 2011*; *Clandinin and Feldheim, 2009*; *Feller and Sun, 2011)*. The fly's compound eye is an assembly of ~750 ommatidia. Each ommatidium contains six outer photoreceptors (R1–R6) that terminate in the first optic neuropil, the lamina; the axons of the central photoreceptors R7 and R8 establish a retinotopic array of terminals in two separate layers of the second optic neuropil, the medulla. In particular, the development of the deepest projecting photoreceptor neuron, subtype R7, has been analyzed in great detail from axon outgrowth to layer-specific targeting and synapse formation (*Hadjieconomou et al., 2011*; *Feller and Sun, 2011*; *Ting, et al., 2007*; *Ting, 2005*). However, to our knowledge, these steps have not yet been shown in the living, developing brain and the underlying types and roles of growth cone dynamics are unknown.

We have recently performed a slow time-lapse intravital imaging study of photoreceptor R1-R6 growth cone dynamics in intact pupae (*Langen et al., 2015*). However, intravital imaging has reduced resolution in deeper brain regions and is limited to early pupal stages. Previous imaging in cultured brains established high-resolution imaging in short developmental time windows (*Medioni et al., 2015*; *Zschätzsch, et al., 2014)*and over long periods at low resolution and with slow time lapse (*Rabinovich et al., 2015*), thus preventing in depth analysis of the role of filopodial dynamics during an entire neural circuit assembly process. Here we present the development of an ex vivo imaging method for *Drosophila* eye-brain development using 2-photon microscopy that allows widely applicable continuous, fast, high-resolution 4D live imaging anywhere in the fly brain throughout pupal development.

We present R7 growth cone imaging at single filopodium resolution for both long periods (up to 24 hr per session) and at high temporal resolution (<1 min), without deleterious effects on normal development. Our measurements show that R7 growth cones do not actively extend after initial target recognition. Concurrently, the vast majority of R7 filopodia are motile and function in growth cone stabilization during layer formation. Loss of the cell adhesion molecule N-Cadherin (*Ting, 2005*; *Lee, et al., 2001*; *Nern, et al., 2008*) reduces filopodial dynamics and causes destabilization of R7 growth cones, resulting in active growth cone 'jumping' between layers even days after targeting has been concluded in wild type. These findings reveal an unexpected role for growth cone filopodia during layer formation and highlight the importance of assessing subcellular dynamics in relation to long-term neuronal development during brain wiring.

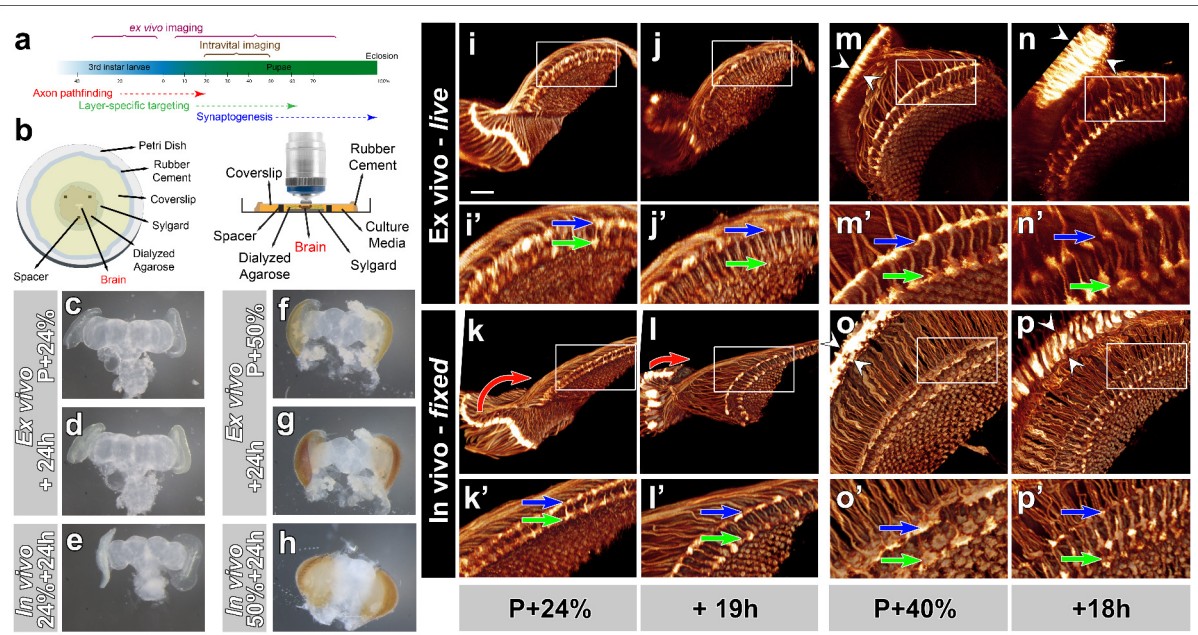

**Figure 1.** Development of Drosophila pupal brains in an imaging chamber. (**a**) Timeline of photoreceptor circuit formation during brain development and the periods accessible by live imaging. (**b**) Ex vivo imaging chamber, top (left) and side (right) views (see *Figure 1—figure supplement 1* for step-by-step assembly). (**c-h**) Changes in brain morphology during development ex vivo v. in vivo. (**c,f**) Pupal brains dissected at P + 24% and P + 50%. (**d, g**) The same brains after 24 hr of development ex vivo. (**e,h**) Brains that were dissected from pupae collected at P + 24% and P + 50% and aged in parallel to the ex vivo brains. See *Figure 1—figure supplement 2* for comparison with free-floating cultures. (**i-p**) Optic lobe development ex vivo v. in vivo (**i'-p'**) magnified details of (**i-p**). All photoreceptors express CD4-tdGFP. Initial layer separation (P + 24% + 19 hr) occurs ex vivo (**i',j'**) similarly to the in vivo controls (**k', l'**) aged in parallel (blue arrows: R8, green arrows: R7). Lamina rotation (red arrows) observed in vivo (**k, l**) is defective ex vivo (**i, j**). Final layer formation and lamina expansion (P + 40% + 18 hr) occurs similarly ex vivo and in vivo, (**m'-n'**) v. (**o'-p'**) (arrows) and (**m-n**) v. (**o-p**) (between arrowheads), respectively. Note that for the ex vivo brains, images of the same specimens were taken at different time points, while for the in vivo controls different brains had to be fixed and imaged for the different time points. Scale bars, 20 μm.

The following figure supplements are available for figure 1:

**Figure supplement 1.** Culture imaging chamber.

**Figure supplement 2.** Brain development in imaging chamber compared to liquid media.

# Results

## Development of ex vivo brain culture in an imaging chamber

In preparation for fast and high-resolution imaging of growth cone dynamics through different phases of brain development (*Figure 1a*) we systematically tested and adapted culture methods of developing *Drosophila* brains (*Gibbs and Truman, 1998*; *Ayaz, et al., 2008*) in an imaging chamber (*Figure 1b*; detailed description in Online Methods and *Figure 1—figure supplement 1*). Pupal eye-brain complexes dissected at P + 24% exhibited only minor overall shape changes after 24 hr culture in the imaging chamber (*Figure 1c, d*). Eye pigmentation became apparent at the end of this time period, indicating developmental progress. In contrast, a parallel control in which the brain developed normally in the pupae during the same time period exhibited a more pronounced expansion of the eye discs, but not yet any obvious eye pigmentation (*Figure 1e*). Similarly, eye-brain complexes dissected at P + 50% and cultured for 24 hr exhibited increasing eye pigmentation, but less overall shape changes than a brain developed inside the fly (*Figure 1f–h*).

To analyze development at the level of axon targeting, we expressed a membrane-tagged GFP (CD4-tdGFP) (*Han et al., 2011*) in all photoreceptor neurons. We again compared the development of a brain in culture (from now on referred to as ex vivo), at different time points with brains that developed normally inside the fly (from now on referred to as in vivo). As shown in *Figure 1i,l* and

*Video 1*, the distance between the terminals of R7 and R8 axons increase significantly after 19 hr ex vivo and in vivo (blue and green arrows). In contrast, a prominent rearrangement of neuropils, where the lamina repositions itself around the medulla, appeared to occur only partially ex vivo (red arrows *Figure 1k–l*). We next compared 18 hr ex vivo and in vivo development of photoreceptor axon projections starting at P + 40%. We observed increases of both lamina and distal medulla thickness (between arrowheads) that occurred both ex vivo and in vivo, suggesting similar developmental progress (*Figure 1m–p* and *Video 1*). These observations suggest that large-scale morphological rearrangement may require factors outside of the eye-brain complex but may have no obvious effects on axon targeting inside the brain.

## Culture and continuous laser scanning do not adversely affect developmental outcome

Next, we sought to determine (1) to what extent the rate of development is affected in culture and (2) the effect of continuous laser scanning during development. To measure developmental speed we compared stage-matched brains ex vivo and in vivo. To assess the effect of laser scanning, we compared a continuously imaged optic lobe (scanned every 30 min for approximately 15 min) with an unscanned control optic lobe of the same brain ex vivo. For brains dissected at P + 22% and cultured for 20 hr we found no difference in distal medulla expansion between the optic lobes subjected to continuous laser scanning and the control optic lobes within the same brains (*Figure 2a, b*). However, this expansion in the ex vivo brains was more rapid than in brains kept in vivo; the latter required 10 hr extra to achieve the same distal medulla size (*Figure 2a, b*). Importantly, the final thickness of the distal medulla was identical and did not increase further in both cases.

Next, we analyzed the development of brains dissected at P + 41% and cultured for 19 hr. Similar to the earlier time point, we found no differences in the levels of distal medulla expansion between the continuously scanned and unscanned optic lobes. In addition, we found no quantitative differences with in vivo controls for medulla expansion at this later stage (*Figure 2d–f*). For the effects of application times of the molting hormone 20-Hydroxyecdysone and usage of a resonant confocal microscope vs a 2-photon microscope see *Figure 2—figure supplement 2*. In sum, at this resolution the developmental outcomes appear normal in culture, and are not affected by continuous 2-photon imaging.

## Imaging at high spatial resolution: Distinct growth cone shapes and filopodia types accompany different developmental stages

We next set out to image R7 growth cone dynamics at high resolution. To visualize individual growth cones we sparsely labeled ~10% of R7 cells, the deepest projecting photoreceptor axons in the *Drosophila* brain (*Fischbach and Dittrich, 1989*) through MARCM (*Lee and Luo, 1999*) using GMR-FLP. The development of R7 axons, particularly with regards to their layer specificity (*Ting, 2005*; *Lee, et al., 2001*; *Clandinin, et al., 2001*) and columnar restriction (*Ting et al., 2007*; *Ting, et al., 2014*) has been studied extensively in fixed preparations.

As before, we compared brains ex vivo and in vivo, which were staged in parallel. We compared cultures starting at P + 20% (*Figure 3a*), P + 40% (*Figure 3b*) and P + 55% (*Figure 3c*) that were imaged continuously for up to 20 hr each (*Video 2*); together these time intervals cover the development from layer selection to synapse formation over a 50 hr time period. As expected, live imaging deep in the brain leads to significant loss of fine structure; it was

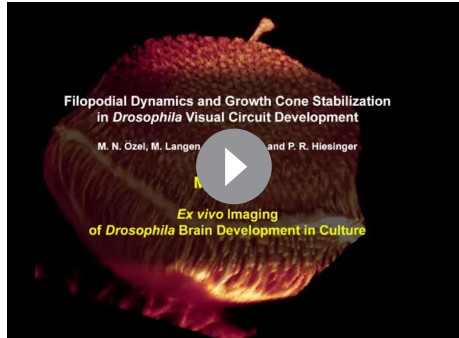

**Video 1.** Ex vivo imaging of *Drosophila* brain development in culture. All photoreceptors are labeled with CD4-tdGFP. Two live imaging sessions (30 min intervals) starting at P + 24% (19 hr) and at P + 40% (18 hr) are shown. Four developmental processes (i) lamina rotation (ii) lamina column expansion (iii) first-stage separation of R7 and R8 terminals and iv) Final layer formation of R7 and R8 terminals, are shown.

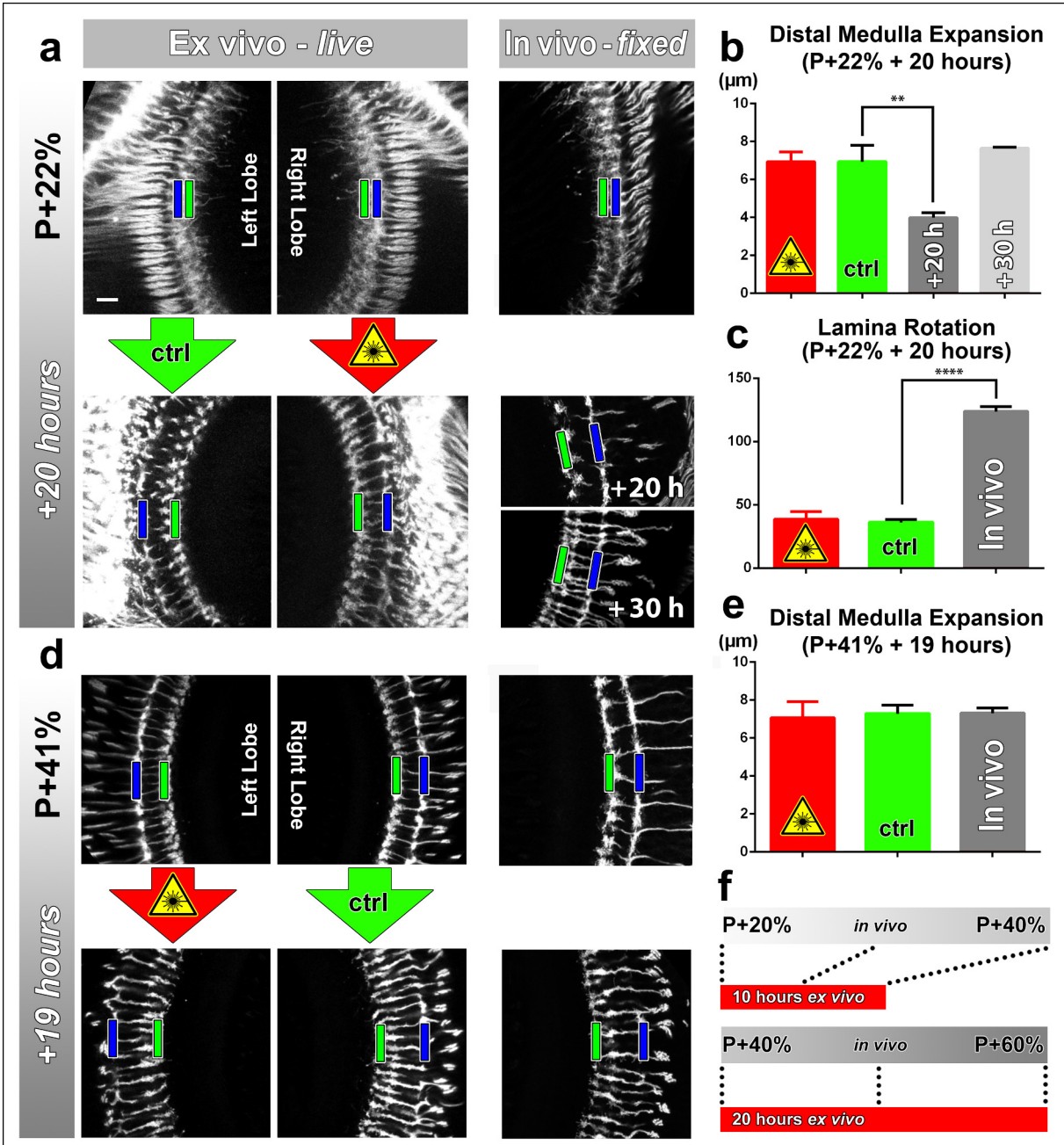

**Figure 2.** Effects of culture conditions and laser scanning on the optic lobe development ex vivo. Two-photon imaging of the medulla was performed with brains cultured at P + 22% for 20 hr (a) and P + 41% for 19 hr (d) all photoreceptors express CD4-tdGFP. For each experiment one image stack was acquired containing both optic lobes of a brain. Next, only one of the lobes was scanned every 30 min. Finally, another stack was acquired with both lobes. Different brains aged in parallel in pupae have been dissected as in vivo controls. (b) Quantification of the layer distance increase in P + 22% cultures. The distance between R8 (green rectangles in a,d) and R7 (blue rectangles) layers increase identically in scanned and unscanned ex vivo lobes, but higher than the in vivo control (p = 0.0036, n = 3). (c) Quantification of the change in the angle between the planes of posterior lamina and the anterior medulla. Ex vivo lobes rotate similarly but slower than in vivo controls (p <0.0001, n = 3). (e) Quantification of the layer distance increase in P + 41% cultures. All groups show a similar increase in the distance between R8 temporary layer and R7 terminals. Error bars depict SEM. (f) Calibration of the developmental speed in culture to in vivo development, based on distal medulla expansion. Scale bars, 10 μm.

The following figure supplements are available for figure 2:

**Figure supplement 1.** Lamina rotation is incomplete ex vivo.

**Figure supplement 2.** Effects of 20-Hydroxyecdysone (20-HE) and type of microscope on imaging in the culture chamber.

difficult to ascertain many faintly labeled filopodia and as a result we consistently counted only about half as many filopodia ex vivo compared to in vivo fixed controls (*Figure 3a–d*). However, this undercount affected filopodia of different lengths equally, resulting in the same mean lengths (*Figure 3e*). Amongst live preparations, a P + 20% preparation after additional 20 hr in culture looked qualitatively and quantitatively identical to fresh preparation at P + 40%; similarly a P + 40% preparation after additional 15 hr in culture looked like a fresh preparation at P + 55%. We conclude that changes in the R7 growth cone structure occur similarly ex vivo and in vivo at this resolution.

The high-resolution structure of R7 growth cones revealed two distinct filopodial 'signatures' before and after P + 50%. Prior to P + 50%, we observed that each R7 growth cone had numerous filopodia that invaded several neighboring columns. During this period R7 growth cones slowly became restricted to their individual columns (*Figure 3a*, lower panel; *Figure 3—figure supplement 1a*). Around P + 40% R7 growth cones underwent extensive loss of filopodia (*Figure 3d–e*; *Figure 3—figure supplement 1a*). Initial high filopodial activity coincides with the beginning of layer separation (*Figure 3a*, upper panel) as lamina neuron axons intercalate between R7 and R8 terminals (*Ting, 2005*). Afterwards, R7 terminals have significantly fewer and shorter filopodia during the remainder of R7/R8 layer selection (P + 45–55%; *Figure 3b*; *Figure 3—figure supplement 1a*).

Surprisingly, longer filopodia reemerge after P + 55% (*Figure 3c, e*). These are fewer in number per growth cone compared to the early-stage filopodia. In addition, these late-stage filopodia often develop bulbous-like tips (*Figure 3c*, arrows) unlike any of the earlier filopodia. These observations

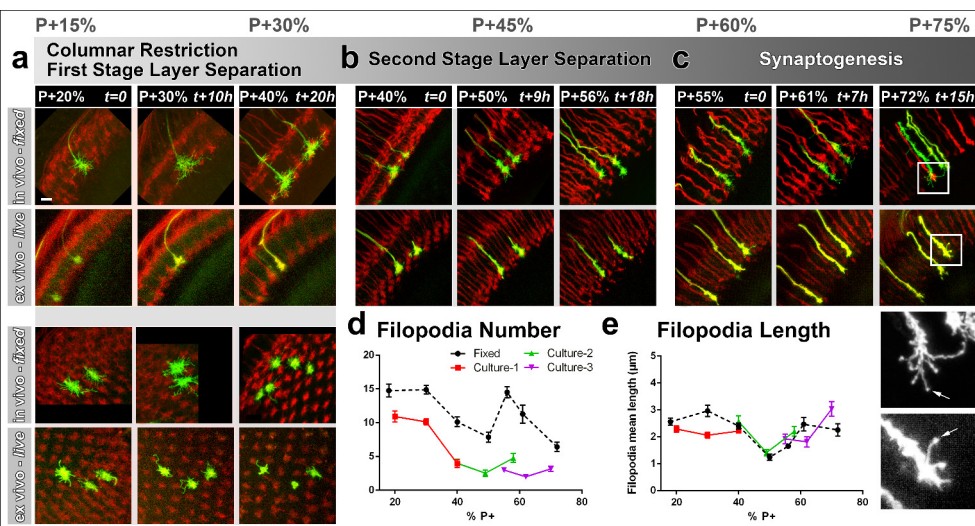

**Figure 3.** Different filopodial signatures accompany separate circuit formation steps. Slow (30 min interval) time-lapse imaging of pupal brains dissected at P + 20% (a), P + 40% (b) and P + 55% (c) in comparison with in vivo fixed controls at the same stages. The same growth cones were analyzed for all live imaging experiments while different samples from parallel aged pupae had to be dissected for the in vivo controls. All photoreceptors were labeled with myr-mRFP and R7 cells were sparsely labeled with CD4-tdGFP using GMR-FLP through MARCM. (a) As the R7 and R8 layers go through their initial separation (upper panel), R7 terminals have numerous filopodia that invade neighboring columns (lower panel), which are pruned around P + 40% both ex vivo and in vivo. (b) As the layers start to reach their final configuration, R7 terminals form a bipartite structure around P + 50%. Filopodia numbers remain low. Around P + 55%, more (shorter) filopodia are observed again as R7 axon assumes a brush-like look. (c) After P + 55% shorter filopodia are pruned and R7 growth cones form new, longer filopodia that are fewer in number and have bulbous tips (arrows). Quantifications of (d) total number of filopodia per growth cone and e, mean length of filopodia through the ex vivo experiments (a-c) and respective in vivo controls. Error bars depict SEM. Scale bars, 5 µm.

The following figure supplement is available for figure 3:

**Figure supplement 1.** Filopodial dynamics are restricted to the growth cone and axon shaft inside the medulla neuropil.

suggest previously undescribed filopodial dynamics that start after P + 55%. During and after this time the R7 terminal undergoes a transition from a morphologically distinct growth cone to an elongated structure with a branched axon shaft, reminiscent of previously observed axonal filopodia in spinal cord culture (*Gallo and Baas, 2011*; *Spillane, et al., 2012*) .However, we only observed filopodia within the medulla neuropil where active layer formation and synapse formation occurs, and not on the main axon leading to the medulla neuropil ( *Figure 3—figure supplement 1*). In summary, distinct growth cone structures accompany separate developmental events and suggest different roles of filopodia during columnar restriction, layer separation and synaptogenesis.

## Imaging at high speed: Most R7 filopodial dynamics are fast, transient and continuous throughout layer formation

To correlate fast filopodial dynamics with developmental events that are hours apart, we applied an imaging protocol that alternated between slow time-lapse imaging of the overall structure and high-resolution fast time-lapse imaging (every 1 min) for 1 hr periods. We focused on critical periods of three major developmental events: the first stage of layer separation until P + 40%, the separation or R7/R8 terminals in what will become the M3 and M6 medulla layers in the adult, and the onset of synaptogenesis.

We found distinct signatures of filopodial dynamics for each of these three processes (*Figure 4a*). Specifically, at P + 28% (during the first stage of layer separation), many transient filopodia (<8 min lifetime) as well as stable filopodia (lifetime >60 min) are apparent (*Figure 4b, c*; *Video 3*). In contrast, at P + 50%, a reduced number of transient filopodia (with the same kinetic characteristics) are present, but no stable filopodia (*Videos 3*, *4*). At P + 60% (onset of synaptogenesis), transient filopodia are dramatically reduced and a new type of stable filopodia has emerged that are less active than those during the first stage of layer separation (*Video 4*). The culture and imaging conditions had little or no deleterious effects, as we observed very similar dynamics for different growth cones that had been in culture for different times at the same developmental timepoints (*Figure 4a*; *Figure 4—figure supplement 1*).

What is the role of transient and stable filopodia during development? Transient filopodia constitute more than 90% of all filopodia at P + 28% and more than 70% of all filopodia at P + 60% (*Figure 4b*). Remarkably, these filopodia exhibit indistinguishable dynamics throughout pupation. We measured no significant changes in their mean lengths (*Figure 4d*), average speed of extension and retraction (*Figure 4e*), levels of inactivity (*Figure 4f*) and the variance of these measurements (blue traces in *Figure 4d–f*). Hence, these dynamics do not correlate with any particular developmental time period or event. Instead, the transient filopodial extensions suggest a continuous function for the even spatial distribution of the columnar and layer structure throughout early developmental stages. The number of transient filopodia reduces increasingly with time (*Figure 4a*) while more cellular processes solidify the adult anatomy.

In contrast to transient filopodia, stable filopodia exhibit a bimodal distribution. A first type of stable filopodia exists up to P + 35% and then rapidly vanishes before P + 50%. These filopodia are significantly longer than transient filopodia, with some exploring up to two columns (*Figure 4b, d*). Surprisingly, their average speed and inactivity (i.e. intervals with no significant extension or retraction) are not significantly different from transient filopodia at this stage (*Figure 4e, f*; *Figure 4—figure supplement 2*). This indicates that at early stages some filopodia appear stable only because they are longer. Unlike the earlier stable filopodia, those that emerge after P + 55% exhibit a significantly

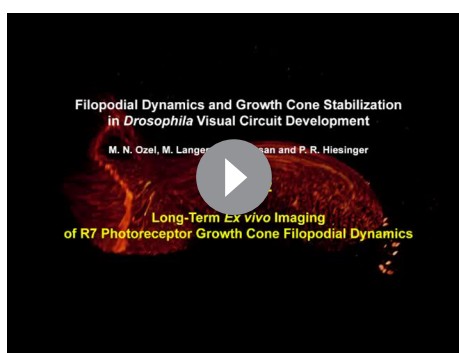

**Video 2.** Long-term ex vivo imaging of R7 photoreceptor growth cone filopodial dynamics. All photoreceptors are labeled with myr-tdTomato and R7 photoreceptors are sparsely labeled with CD4-tdGFP using GMR-FLP. Three live imaging sessions (30 min intervals) starting at P + 22% (21 hr), P + 42% (19 hr) and P + 55% (15 hr) are shown. The development of the filopodial structure of R7 growth cones are shown throughout layer and synapse formation.

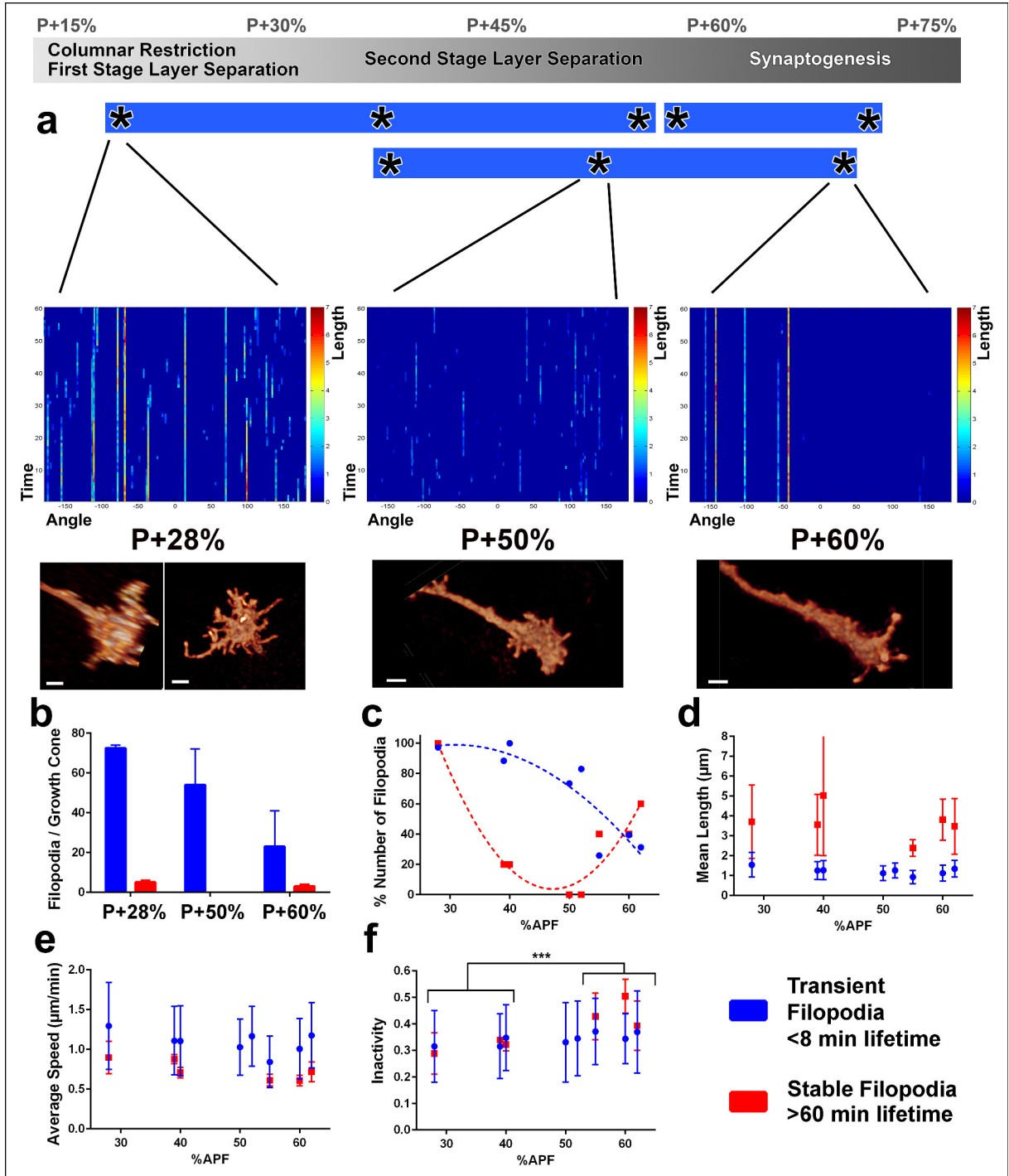

**Figure 4.** Distinct classes of transient and stable filopodia underlie different developmental events. Fast (1 min interval) time-lapse imaging was performed at multiple points of three ex vivo experiments. (a) Three time points are shown; during the first-stage (P + 28%) and second-stage (P + 50%) layer formation, and synaptogenesis (P + 60%). 3D graphs (upper panel) show the dynamics of individual filopodia observed in a one hour period. In the heat maps on blue background, individual filopodia are shown as verticals lines. The filopodia were sorted by their initial orientation angle (x-axis). The length of the vertical lines represents the life time of the filopodia (time on the y-axis). The color map indicates the length (μm) for each filopodium through time. Representative images of the growth cones at the above time-points (lower panel). See *Figure 4—figure supplement 1* for heat maps and representative images at all time points. (b) Numbers of filopodia per growth cone for the time-points shown in a; for filopodia with lifetime <8 min (transient) and lifetime >1 hr (stable). (c) Numbers of filopodia relative to the numbers at P + 28% for all time-points imaged. Fitted curves: y = 28.17 + 4.597x-0.075$x^2$ (transient) and y = 583.1–24.53x + 0.26$x^2$ (stable). (d) Mean length (μm) (e) Average speed (μm/min) and (f), Inactivity (ratio of intervals with no significant extension or retraction) for transient and stable filopodia at all time-points. Stable filopodia observed after P + 50% have significantly

*Figure 4 continued on next page*

*Figure 4 continued*

higher inactivity than those observed before (Means: 0.3002 v. 0.4346, p = 0.0002, n = 14 for each). See *Figure 4—figure supplement 2* for these parameters as a function of filopodia lifetime on the same growth cone. Error bars depict SD. Scale bars, 2 μm.

The following figure supplements are available for figure 4:

**Figure supplement 1.** Fast filopodial dynamics throughout pupal development.

**Figure supplement 2.** Filopodial dynamics as a function of lifetime.

greater inactivity compared to both the transient filopodia at this stage as well as the filopodia in earlier stages (*Figure 4f*; *Figure 4—figure supplement 2*). Combined with their peculiar bulbous tips (*Figure 3c*), they define a distinct class of filopodia in both structure and dynamics. The two types of stable filopodia correlate with different developmental subprograms: The first type accompanies columnar stabilization and restriction of the growth cone, while more layers are being formed. In contrast, the second type of stable filopodia emerges after layers are defined and interactions with presumptive synaptic partners commence. The time period around P + 50% where stable filopodia are absent matches precisely the moment when the final R7 and R8 layers are defined.

In summary, measurements of fast filopodial dynamics reveal that the majority of R7 filopodia are transient and may function during continuous column and layer stabilization; in contrast, distinct classes of stable filopodia may be substrates for the specific types of neurite interactions underlying the developmental events they accompany.

## Single growth cone tracking reveals continuous R7 growth cone stabilization during layer formation

The temporary absence of stable filopodia around P + 50% marks a critical developmental period when the final R7 and R8 layers are determined (*Ting, 2005*). In this well-studied model for layer formation the R8 growth cone is known to actively extend (*Timofeev et al., 2012*). In contrast, how R7 reaches its final target layer is less clear. R7 ends up in the deepest layer of the distal medulla through one of two processes: it may extend away from a temporary layer to a new, more proximal

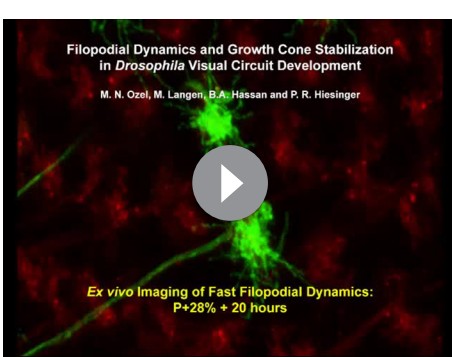

**Video 3.** Ex vivo imaging of fast filopodial dynamics-1. All photoreceptors are labeled with myr-mRFP and R7 photoreceptors are sparsely labeled with CD4-tdGFP using GMR-FLP. Live imaging started P + 28% and continued for 20 hr. We used an alternating slow (30min intervals) imaging of the general structure and fast (1min interval) imaging of two growth cones at higher resolution at three different time points. Fast filopodial dynamics of the same two growth cones at P + 28%, P + 40% and P + 55% are shown.

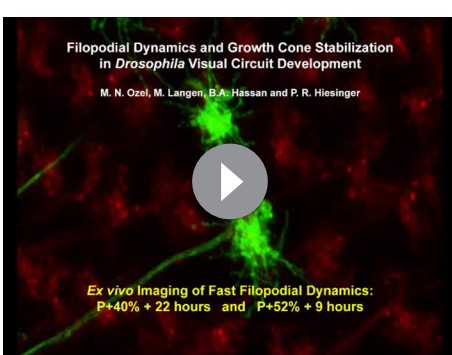

**Video 4.** Ex vivo imaging of fast filopodial dynamics-2. All photoreceptors are labeled with myr-mRFP and R7 photoreceptors are sparsely labeled with CD4-tdGFP using GMR-FLP. Two live imaging sessions starting at P + 40% (22 hr) and P + 52% (9 hr) are shown. We used an alternating slow (30 min intervals) imaging of the general structure and fast (1 min interval) imaging of two growth cones at higher resolution at different time points. Fast filopodial dynamics of the same two growth cones at P + 40%, P + 50% and P + 60% and fast filopodial dynamics of another three growth cones at P + 52% and P + 62% are shown.

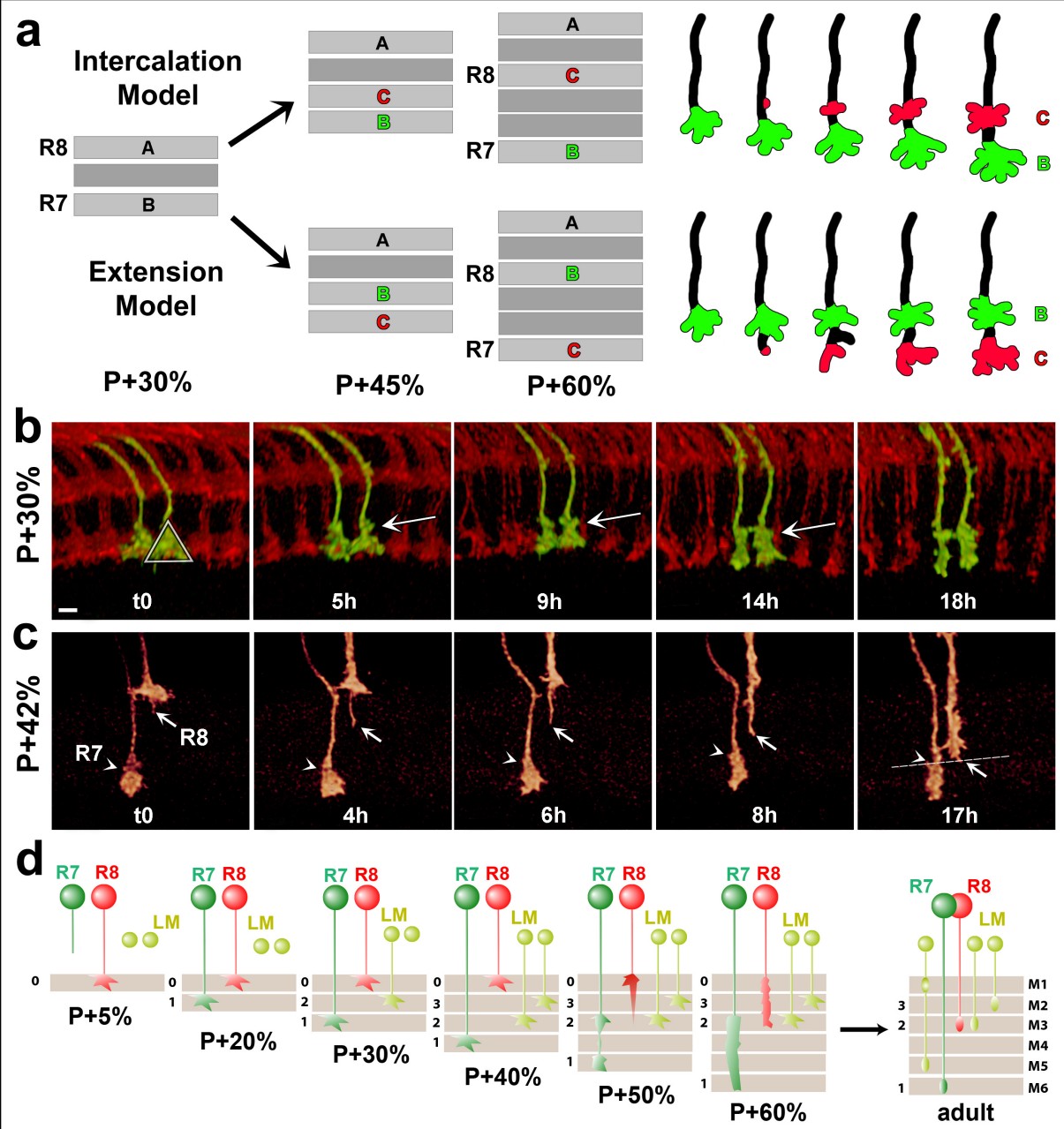

**Figure 5.** R7 growth cones do not actively extend in the medulla. (**a**) R7 may reach its final target layer through active extension or passive displacement and intercalation. (**b**) Live imaging starting at P + 30%. All photoreceptors were labeled with myr-mRFP and R7 cells were sparsely labeled with CD4-tdGFP using GMR-FLP through MARCM. R7 growth cone (triangle) initially has a cone structure. As the layer formation progresses, a new varicosity (arrow) is formed from the axon shaft. This structure expands further and by P + 50% the entire terminal thickens. See *Figure 5—figure supplement 1* for all time points. (N = 31). (**c**) Live imaging starting at P + 42%. Both R7 and R8 cells were sparsely labeled with CD4-tdGFP using hsFLP. R7 axon has already formed its distal varicosity (arrowhead); the R8 axon has extended a single filopodia proximally (arrow). Later, this filopodia reaches to the R8 final layer and forms the new terminal. R7 terminal shows no active extension activity. (N = 17 for R7 and 15 for R8). (**d**) Model of layer formation in the distal medulla. After their arrival to the medulla R7 and R8 terminals are initially separated by intercalation of lamina cell (LM) axons. After P + 40%, R8 growth cones actively extend to new layer while R7s remain in their arrival layer throughout. Scale bars, 3 μm.

The following figure supplement is available for figure 5:

**Figure supplement 1.** Single growth cone tracking demonstrates R7 terminals remain passive throughout layer formation without a stationary landmark.

layer (active model) (*Hadjieconomou et al., 2011*; *Feller and Sun, 2011*; *Mast, et al., 2006*) alternatively, more layers are intercalated by other neurons while R7 remains in the same layer throughout (*Ting, 2005*) (passive model) (*Figure 5a*). The recent finding that R7 is in close proximity with its Dm8 target dendrites as early as P + 17% (*Ting et al., 2014*) supports the passive model (*Ting, 2005*); however, it remains unclear whether R7 growth cones actively participate in any part of the layer formation process. Live imaging of the entire process of layer formation can provide an unequivocal answer to the question whether R7 growth cones exhibit any extension activity by following the same growth cones over time.

We used a time-lapse interval of 30 min to track individual growth cones and their shape changes (*Video 5*). At P + 30% the R7 growth cone exhibits a cone-shape that expands towards the terminal ending from its thin axonal process (triangle in *Figure 5b* = 0). Over the next 18 hr we observed a gradual change of this shape, but no extension away from it (*Figure 5b*). How does the R7 axon accommodate new layer formation in the expanding distal medulla without extension? As shown in *Figure 5b* new varicosity emerges distally on the axon shaft of the cone-shaped growth cone (t = 5 hr, arrow). This varicosity expands to give the entire terminal a bipartite structure (t = 5–14 hr); these observations suggest the intercalation of a new layer and support the passive model. Importantly, the continuous observation of the same growth cone and its dynamics (*Video 5*, *Figure 5—figure supplement 1*) unequivocally reveals the lack of active extension without the need for a stationary landmark that is necessary in fixed images to verify movement.

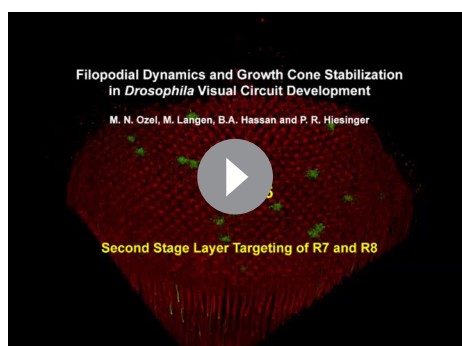

**Video 5.** Second stage layer targeting of R7 and R8. Two live imaging experiments are shown. (1) All photoreceptors are labeled with myr-mRFP and R7 photoreceptors are sparsely labeled with CD4-tdGFP using GMR-FLP. Imaging started at P + 30% and continued for 18 hr, with 30 min intervals. Two R7 growth cone tips (red arrow) were followed. At the 2.5 hr mark a varicosity starts to develop from the axon shaft and expands over the next 15 hr, contributing to the elongation of the R7 axon. Note that being able to follow the same growth cone tip based on its unique filopodial structure allows us to verify lack of active extension without a stationary landmark. (2) R7 and R8 photoreceptors were sparsely labeled with CD4-tdGP using hs-FLP. Imaging started at P42% and continued for 21 hr. We used an alternating slow (30 min intervals) imaging of the general structure and fast (1 min interval) imaging of two neighboring R7 and R8 growth cones at higher resolution at different time points. R8 axon relocates to its final layer by sending a single filopodia proximally, which is initially very dynamic but later stabilizes and expands in the new layer, forming the new R8 terminal. In contrast, R7 terminal elongates along the axon shaft, but no directed extension activity is observed on the growth cone.

In contrast to R7, the growth cones of R8 extend a single filopodium towards their final layer (*Figure 5c*, arrow); this filopodium is initially highly dynamic and exhibits almost complete retractions and re-extensions. It is finally stabilized in a deeper layer and gradually becomes thicker to form the new R8 terminal in the same layer as the distal end of the intercalated R7 varicosity (*Figure 5c*, arrowhead). The formation of the bipartite R7 growth cone and its intercalating varicosity precedes the stabilization of the dynamically extending and retracting R8 process. This observation suggests that some other cell type first defines the layer where first the R7 growth cone forms its expanding varicosity and finally R8 targets.

In summary, our live data demonstrate that R7 terminals do not actively extend after P + 30%. Instead, R7 growth cones arrive directly to their final layer and are only passively dislocated by the intercalation of other axons and dendrites (*Figure 5d*). This process requires their continuous stabilization.

## N-Cadherin is required for growth cone stabilization, but not for layer-specific targeting

The idea of passive retention implies that R7 growth cones do not engage in any active targeting process after P + 30% and is consistent with the continuous transient filopodial dynamics shown above (*Figure 4*). However, previous mutant analyses described R7 targeting defects into layers that form after P + 30%. For example, the homophilic adhesion molecule N-Cadherin (CadN) (*Hatta et al., 1985*) has emerged as a major regulator of synaptic layer specificity and

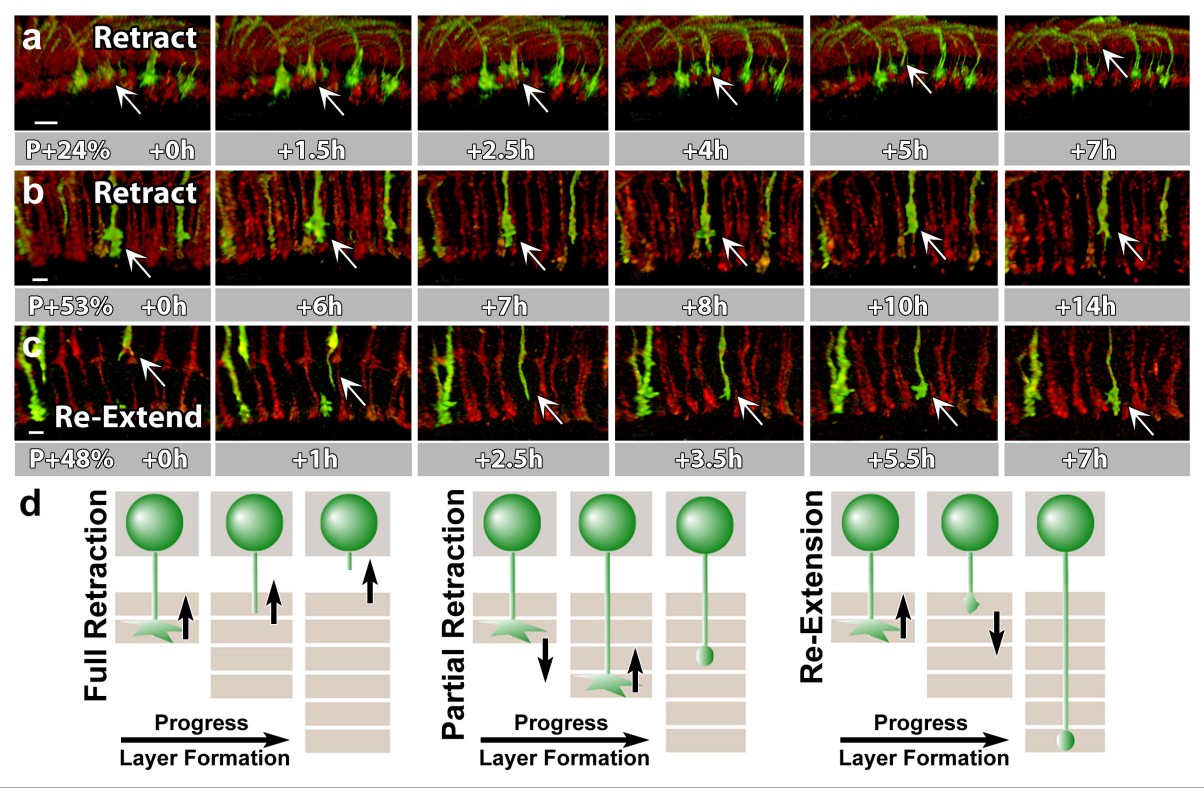

**Figure 6.** N-Cadherin is required for the stabilization but not the layer specific targeting of R7 growth cones. All photoreceptors were labeled with myr-mRFP. CadN[405] R7 cells were generated with MARCM, using GMR-FLP and positively labeled with CD4-tdGFP. (**a**) Live imaging started at P + 24% shows a mutant R7 growth cone (arrow) that retracts from its target layer over the course of 5 hr. (**b**), Live imaging started at P + 53% shows a mutant R7 growth cone (arrow) that retracts from its target layer over the course of 10 hr. Some mutant axons retract completely from the medulla (**Figure 6—figure supplement 1**) (**c**) Live imaging started at P + 48% shows an R7 axon (arrow) that has been retracted to the edge of distal medulla but re-extends and attempts to re-innervate both wrong (5.5 hr) and the right (7 hr) layers. (**d**) Schematics of observed retraction and re-extensions events. Left and middle: Full Retraction leads to complete loss of the R7 axons from the medulla (left), while partial retraction (middle) leads to R7 terminals in an incorrect layer. Number of mislocalized terminals: 33% (n = 85) at P + 40% and 56% (n = 62) at P + 52%. Right: Previously retracted R7 axons can re-extend, even days after they would have been stabilized in wild type. 52% (n = 23) of retracted axons at P + 40% re-extended before P + 50%Scale bars, 5 µm.

The following figure supplement is available for figure 6:

**Figure supplement 1.** *CadN* mutant R7 axons may retract completely from the medulla.

its loss of function causes R7 mistargeting to the R8 layer (*Ting, 2005*; *Lee, et al., 2001*). Previous studies focused on structure-function analyses of CadN (*Nern et al., 2005*) and its molecular interactions with other proteins (*Prakash et al., 2009*) and found that the penetrance of the mistargeting defect increased over time, suggesting retractions (*Ting, 2005*). However, how loss of *CadN* causes mistargeting or retractions is still unclear. In particular, it is unknown what changes in the growth cone dynamics cause this phenotype.

To investigate these aspects we performed live imaging of *CadN* mutant R7 axons (positively labeled using MARCM) in an otherwise wild-type brain (*Video 6*). We observed that almost all R7 terminals arborized correctly in a layer right below R8 terminals upon arrival at the medulla prior to P + 20%. At P + 23%, some of the 'oldest' mutant terminals that first arrived at the medulla were mislocalized (17% of all R7 terminals, n = 54). As predicted (*Ting, 2005*), this increase is due to the retraction of R7 terminals which were initially in the correct position (*Figure 6a, d*). These retractions were always preceded by a gradual collapse of their filopodial structure that could predict the remobilization of the growth cone at least 2 and up to 10 hr prior to retraction.

The fraction of mislocalized terminals increased to 33% (n = 85) by P + 40% and 56% (n = 62) by P + 52%. In addition, these numbers are underestimates since some of the mutant terminals retract completely from the medulla (*Figure 6—figure supplement 1*). Retractions continued even after the wild-type neurons formed their final layers (*Figure 6b, d*), resulting in the previously observed penetrance of 70% in adult brains (*Lee et al., 2001*). These late retractions could be the consequence of dying back axons, or, alternatively, *CadN* deficiency is sufficient for R7 axons to regain active mobility days after their targeting is concluded.

Live observations of growth cone dynamics provided a clear distinction of these two possibilities: we observed that 52% (n = 23) of retracted axons at P + 40% APF actually *re-extended* towards more proximal layers within the next 8 hr. These axons often re-arborize in both correct and incorrect layers (*Figure 6c, d*), but again fail to stabilize those arborizations (*Video 6*). This phenotype was previously impossible to recognize in fixed preparations and masked by the overall increase in mistargeting penetrance. These data show that *CadN* mutant axons regain motility for days after their targeting should have been concluded. We conclude that CadN is not required for targeting *per se*, but for the stabilization of R7 growth cones after initial targeting.

## N-Cadherin is required for fast filopodial dynamics

What is the role of filopodia in growth cone stabilization? Our R7 filopodial dynamics measurements revealed that >90% of all filopodia were transient and exhibit continuous, stochastic extension/retraction dynamics that did not correlate with any specific developmental processes (*Figure 4*). These dynamics are consistent with continuous stabilization of the passively retained R7 growth cones throughout development (*Figure 5*). If filopodia control growth cone stabilization, then CadN growth cones should exhibit reduced filopodial dynamics.

*CadN* R7 growth cones do not appear obviously disrupted as long as they remain in their initial, correct arrival layer (*Figure 6*) and filopodia numbers are not significantly affected at P + 28% (*Figure 7a*). However, both transient and stable filopodia of mutant growth cones exhibit reduced average speed of extension/retraction (*Figure 7b*, *Video 7*). As a consequence, both types of filopodia are on average also significantly shorter than wild-type (*Figure 7c, d*). These findings point to a general slow-down of the filopodial dynamics in *CadN* growth cones and suggest that N-Cadherin mediated adhesion (*Hatta et al., 1985*) is important for the stabilization of R7 growth cones through filopodial interactions at the target layer.

In summary, we find that filopodial dynamics predict growth cone stabilization in a specific layer. This attachment of the growth cone in a specific layer is a continued requirement long after initial targeting is completed and it is further reflected in the majority of transient filopodial extension/retraction dynamics. Loss of *CadN* reduces these dynamics and increases the likelihood of layer destabilization even days after targeting is concluded.

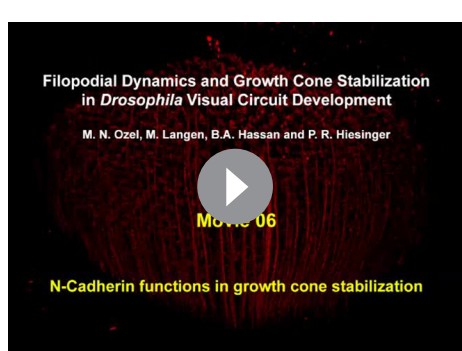

**Video 6.** N-Cadherin functions in growth cone stabilization. All photoreceptors are labeled with myr-mRFP and approximately 10% of R7 photoreceptors were made mutant for *CadN* and labeled with CD4-tdGFP using GMR-FLP through MARCM. Three live imaging sessions are shown. (1) Starting at P + 24% (17 hr). R7 axons arrive correctly to their target layer but they gradually retract from it, preceded by growth cone collapse. (2) Starting at P + 53% (20 hr). Retractions continue despite the wild-type photoreceptors reached their final layer configurations. (3) Starting at P + 42% (11 hr). Some of the R7 axons that retracted at the earlier stages re-extend back into the distal medulla. Note that the growth cones are streamlined during active movement but show expansion while the axons attempt to re-innervate various medulla layers.

## Discussion

Fast filopodial movements of growth cones are thought to play important roles during brain development, but their types and roles remain largely unknown. In this study we developed a brain culture live imaging system that is applicable for all developmental stages of *Drosophila* brain development across a wide range of temporal and spatial scales. We used this system to

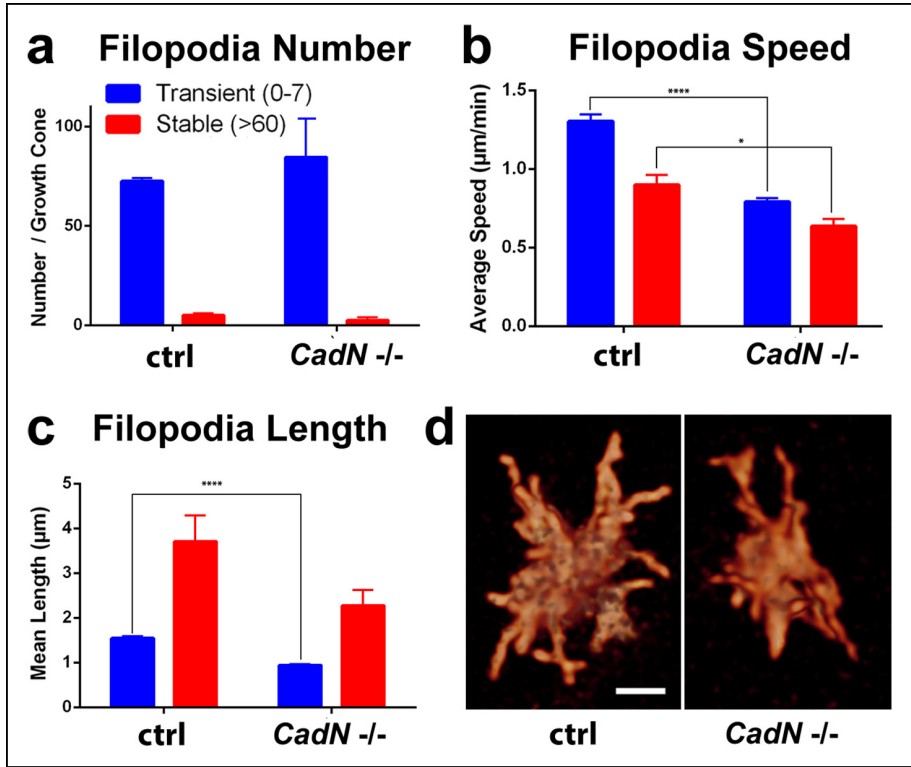

**Figure 7.** N-Cadherin is required for fast filopodial dynamics. CadN[405] R7 cells were generated with MARCM, using GMR-FLP and positively labeled with CD4-tdGFP. Fast (1 min interval) time-lapse imaging was performed at P + 28%. (**a**), The average numbers of filopodia per growth cone are not significantly different between wt and CadN[405]. (**b**) Mutant filopodia are slower, (for transient, means wt: 1.303 (n = 143), CadN[405]: 0.791 (n = 169), p<0.0001; for stable, means wt: 0.898 (n = 10), CadN[405]: 0.636 (n = 5) p = 0.0199) and (**c**) shorter (for transient, means wt: 1.542 (n = 143), CadN[405]: 0.939 (n = 169), p <0.0001; for stable: means wt: 3.707 (n = 10), CadN[405]: 2.275 (n = 5), p = 0.1257). (**d**) CadN[405] R7 growth cones at the correct layer. Scale bars, 5 μm.

investigate the role of R7 growth cone dynamics during layer formation throughout a 3-day developmental period. Our findings provide new insights into a major role of filopodia during column- and layer-specific growth cone stabilization. In addition, these observations indicate that growth cone dynamics can influence axon terminal targeting without a direct role in specifying the target and more generally may implement simple rules during brain wiring.

## Ex vivo live imaging of *Drosophila* brain development at the resolution of filopodial dynamics

Our ex vivo brain development system in a closed imaging chamber allows continuous laser scanning during development for at least 20 hr per session. For longer imaging periods, the system can be modified to a semi-open state with perfusion (*Williamson and Hiesinger, 2010*). However, the ease of the closed chamber outweighed the advantage of longer imaging periods in our hands. Our key goal was to follow subcellular dynamics at the resolution limit of conventional light microscopy with fast enough time-lapse to quantitatively describe subcellular dynamic properties in developing brains over many hr. Important advances in Drosophila ex vivo brain imaging have recently established high-resolution imaging in short developmental time windows (*Medioni et al., 2015*; *Zschätzsch, et al., 2014)* and over long periods at low resolution and with slow time lapse (*Rabinovich et al., 2015*). We identify phototoxicity and drift as key problems to obtain high spatial and temporal resolution 3D dynamics data over long developmental time periods, for which the imaging system presented here provides a successful approach.

We have tested our system for developmental processes ranging from L3 brain development (cell migration, data not shown) and throughout pupal development (growth cone dynamics). We further

provide the calibration of developmental progress in this culture system under imaging conditions. For example, morphological changes of the eye and 'lamina rotation' occur only incompletely outside of the fly's head (*Figure 2c* and *Figure 2—figure supplement 1*). In contrast, early layer formation of photoreceptor axonal projections are accelerated with normal outcome; development after P + 40% occurs with identical speed in our ex vivo system and in vivo. These findings indicate that layer and synapse formation are not directly dependent on distal tissue morphogenesis. However, different developmental processes must be calibrated for their ex vivo progress compared to in vivo development in the fly. Based on our quantitative analyses of layer formation in the distal medulla, we anticipate that the developmental progress of more proximal brain regions will be similar to the calibrated optic lobe development.

We show that conventional 2-photon microscopy can safely be used over long periods with virtually no drift and at high resolution in our imaging chamber when following a simple 'no bleaching' rule. In some cases we even observed mild photobleaching (e.g. *Figure 2d*) without adverse effects on developmental progress. We conclude that as long as there is no significant decrease in the signal intensity over time, 2-photon imaging *per se* does not negatively affect the development. In addition, ex vivo imaging has the advantage that the culture media allow pharmacological manipulations which are not easily possible in vivo or with intravital imaging.

In summary, the ex vivo imaging system and conditions developed here allow to observe live the formation of neural circuits anywhere in the *Drosophila* brain. Importantly, imaging at different spatial and temporal scales allows relating fast, high-resolution filopodial dynamics to much slower, long-term developmental processes.

## Linking fast dynamics to long-term development: The role of filopodia

Growth cone behavior is highly dynamic and context-dependent (*Mason and Erskine, 2000*). Understanding the role of growth cone dynamics as part of a longer developmental process requires observation in their normal environment. Growth cone filopodia have traditionally been interpreted as probes that detect guidance cue gradients (*Gallo and Letourneau, 2004*; *Zheng, et al., 1996*) or as 'sticky fingers' that provide the traction required for growth cone migration (*Heidemann, 1990*; *Chan and Odde, 2008*). Our characterization of the R7 growth cones revealed a different role for the vast majority of its filopodia during layer formation in the distal medulla: Surprisingly, more than 90% of R7 filopodia exhibit apparently stochastic extension/retraction dynamics that do not correlate with any major structural change during layer formation in the distal medulla. Instead, these movements are fast, transient and only slowly reduce over the period of days during brain development, while new neurons innervate and new layers form.

What is the role of these filopodia? Our imaging data revealed that R7 growth cones do not actively extend after their initial target recognition, in contrast to some of the earlier models (*Hadjieconomou, et al., 2011*; *Clandinin and Feldheim, 2009*; *Ting and Lee, 2007*). Instead, other axons and dendrites intercalate while R7 growth cones define the most proximal boundary of the distal medulla. Hence, R7 must stably maintain their position while active intercalation of other neurons, e.g. R8 extension, pushes the R7 layer proximally (*Figure 5d*). This stabilization is consistent with continued filopodial extension/retraction dynamics that are decreasingly required as the final adult column and layer organization solidifies. However, we note that our imaging data do not establish a causal relationship between the observed dynamic behaviors.

The stabilizing function is reminiscent of zebrafish retinotectal axons which display a broadened structure while resting but are more streamlined during extension (*Kaethner and Stuermer, 1992*). Stabilization through

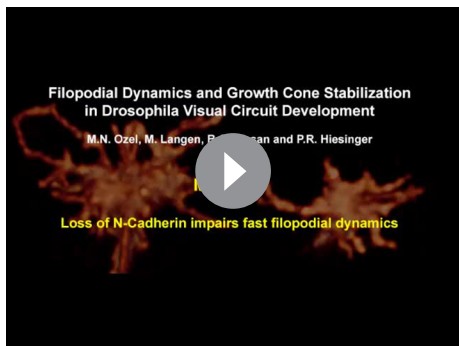

**Video 7.** Loss of N-Cadherin leads to reduced filopodial dynamics. Representative wild type and *cadN* mutant R7 growth cones are shown at P + 28%. Extraction of individual filopodia reveals reduced dynamics over the same time period (1 hr with 1 min time lapse) as shown in the quantifications in *Figure 7*.

filopodial dynamics is further supported by the observation of 'jumping' growth cones in CadN mutants, as discussed below. Finally, we also observed a previously undescribed kind of filopodia that emerge at later stages. These are more stable, appear to coincide with the timing of synaptogenesis and are reminiscent of densities observed in hippocampal cultures using VAMP-GFP (*Smith et al., 2000*). However, adult R7 synapses are restricted to the main axonal trunk of the R7 terminal and the precise roles of these late, stable filopodia away from the main axonal trunk remain to be investigated.

Our observation of growth cones from initial arrival through layer formation and finally synaptogenesis reveals a remarkable transitioning of the R7 terminal shape from a more classical growth cone to an elongated structure with branched axon shaft (*Figure 3—figure supplement 1a*). Filopodia on this extended R7 axon are reminiscent of axonal filopodia observed in spinal cord culture (*Gallo and Baas, 2011*; *Spillane, et al., 2012*), but restricted to the axon shaft inside the medulla neuropil where layer formation and synaptogenesis occur (*Figure 3—figure supplement 1b, c*). Before P + 40%, we only observe filopodia at the axon tips as in classical growth cones. Therefore, we suggest these structures are still growth cones although they appear to use filopodia as a means to stabilize rather than as a substrate for migration. We think that the subsequent transition morphologies could reasonably be interpreted as an extended growth cone or a distinct and short part of the proximal axon that got recruited to new active functions during layer formation (*Figure 3—figure supplement 1a*).

## The role of N-Cadherin: Stabilizing the targeting decision, rather than making it

Our findings support a model in which continuous stabilization of R7 growth cones in a column/layer grid depends on the levels of N-Cadherin (CadN). The observation of mistargeted photoreceptor axons (*Lee et al., 2001*) as well as its classical role in axon guidance (*Matsunaga et al., 1988*) have previously led to the interpretation of CadN as a guidance cue. The interpretation as a part of a specificity code is complicated by the observation that CadN is expressed in all presynaptic and postsynaptic neurons during distal medulla development; however, temporal regulation (*Petrovic and Hummel, 2008*) as well fine-tuning of expression levels (*Schwabe et al., 2014*) have been proposed as solutions.

Our live imaging data reveal that CadN-deficient R7 axons have no initial targeting defects and CadN does not function as a target layer-specific cue. Instead, growth cones fail to stabilize and engage in an aberrant process of 'jumping' between incorrect and correct layers. Remarkably, CadN-deficient R7 growth cones retain the ability to jump between distal medulla layers for days after their normal targeting should have been concluded and stabilized, presumably through the filopodial dynamics described here. CadN has been shown to localize to the filopodia of R1-6 photoreceptor axons in the lamina neurpil (*Schwabe et al., 2013*); we speculate that R7 growth cones could use the surface area of their filopodia to form stabilizing adhesions through CadN. Consistent with this interpretation, loss of CadN reduces filopodial extension/retraction dynamics and 'jumps' between medulla layers are preceded by a slow, several hours-long filopodial retraction process.

CadN mediated adhesive interactions were shown to be essential for growth cone migration in primary neuronal cultures (*Bard et al., 2008*). We observed decreased filopodial lengths in CadN growth cones, consistent with the longer neurite lengths observed with increased CadN mediated adhesions. While such interactions were previously correlated with growth cone velocity, our observations provide a link to their stability in a different context.

CadN-mediated adhesion with other medulla cells is required for R7 terminals to end up in the correct layer independent of its signaling function (*Yonekura et al., 2007*). This finding is consistent with a growth cone stabilization function via interactions with many different medulla cells, independent of the correct synaptic partners. This idea is further consistent with the widespread expression of CadN in many cell types. In sum, our data together with previous observations support a 'non-guidance cue' function for CadN in stabilizing the positions and contacts of neurites once the targeting is complete. Indeed, several cell adhesion molecules previously thought to function as guidance cues have recently been shown to exert 'non-cue' functions cell-autonomously (*Petrovic and Schmucker, 2015*) and through implementing simple developmental rules (*Hassan and Hiesinger, 2015*). Initial R7 targeting to the correct layer could be achieved by other molecules or by a developmental rule such as *'stop at the first target layer encountered past the pioneer R8 axon, and then*

*stabilize'*. It is possible that such a rule could result in the correct initial targeting of R7 and L-cell axons simply by their arrival order (*Figure 5d*), requiring no layer-specific molecular code (*Hassan and Hiesinger, 2015*).

## Materials and methods

### Genetics

Pan-photoreceptor labeling was done by GMR-Gal4 expressing the membrane tethered CD4-tdGFP (*Han et al., 2011*). Sparse R7 labeling as well as the generation of *CadN* mutant R7 neurons were achieved through MARCM (Mosaic analysis with a repressible cell marker) (*Lee and Luo, 1999*) using GMRFLP. GMR-myr-RFP or GMR-myr-tdTomato was used to label all photoreceptors in the background.

Fly stocks: i);; *GMR-Gal4, UAS-CD4-tdGFP* ii) *GMR-FLP; GMR-Gal4; FRT80B, UAS-CD4-tdGFP* iii) *GMR-myr-RFP;; FRT80B, tub-Gal80* iv) *GMRFLP; FRT40A, tub-Gal80; GMR-Gal4, UAS-CD4-tdGFP, GMR-myr-RFP* v); *FRT40A, CadN405*; vi) *hs-FLP; GMR-FRT-w + -FRT-Gal4; UAS-CD4-tdGFP* vii) *GMR-myr-tdTomato; FRT80B, tub-Gal80*.

### Histology and fixed imaging

Eye-brain complexes were dissected in PBS, fixed in 3.7% paraformaldehyde (PFA) in PBS for 40 min, washed in PBST (0.4% Triton-X) and mounted in Vectashield (Vector Laboratories, CA). Images were collected using a Leica TCS SP5 confocal microscope with a 63X glycerol objective (NA = 1.3).

### Brain culture

The culture chambers were built inside 60x15 mm petri dish lids with a layer of Sylgard 184 (Dow Corning) at the center (2 cm in diameter). 200 μm thick X-ray films cut in 1x1 mm pieces were used as spacers to prevent the coverslip from crushing the tissue. 2% low melting point agarose was prepared in water and dialyzed in pure water for 48 hr with changing the water every 12 hr at room temperature, then stored at 4°C.

The culture media was modified from a previous recipe (*Ayaz et al., 2008*). It was prepared with 1:10 fetal bovine serum (FBS), 10 μg/ml human insulin recombinant zinc (Stock: 4 mg/ml), 1:100 Penicillin/streptomycin (Stock: 10000 IU/ml penicillin, 10 mg/ml streptomycin), 1 μg/ml 20-Hydroxyecdysone (Stock: 1 mg/ml in ethanol) in Schneider's Drosophila Medium. All were acquired from Life Technologies. Brains were dissected in chilled Schneider's Medium and mounted in 0.4% dialyzed low-melting agarose diluted in the culture media. Step-by-step chamber assembly (*Figure 1—figure supplement 1*):

1. Oxygenize culture medium at room temperature.
2. Melt a piece of 2% dialyzed agarose. Mix with culture media preheated to 42°C at 1:4 ratio (to the final concentration of 0.4%). Keep the mixture at 32°C.
3. Dissect brains in chilled Schneider's Drosophila Medium. Keep them in chilled culture medium until mounting.
4. Place the brain (with a pipette) at the center of Sylgard layer in a 30-40 μl drop of the diluted agarose ( *Figure 1—figure supplement 1a (ii-iii)*).
5. After correctly positioning the brain, place a coverslip (circular, 4 cm diameter) on the drop ( *Figure 1—figure supplement 1a (iv)*).
6. Glue the coverslip to the petri dish at 4 points using rubber cement.
7. After the polymerization of agarose (15-20 min), fill the rest of the space between coverslip and the petri dish with culture media (Figure1-Supplement 1a (v)).
8. Seal the chamber completely with rubber cement (Figure1-Supplement 1a (vi)).

The final imaging chamber ( *Figure 1—figure supplement 1b,c*) provides sufficient oxygen and nutrients through diffusion for at least 24 hr. 20-Hydroxyecdysone is excluded from the cultures that start after 50% APF. This is due to previously measured physiological titers (*Paul Bainbridge and Bownes, 1988*) as well as our experimental data (*Figure 2—figure supplement 2a–h*).

## Live imaging

Live imaging was performed at room temperature using a Zeiss LSM 780 multiphoton microscope with a 40X LD water objective (NA = 1.1) or a Leica SP8 MP microcope with a 40X IRAPO water objective (NA = 1.1) with a Chameleon Ti:Sapphire laser (Coherent). For single-channel CD4-tdGFP imaging, excitation was done at 900 nm. For double-channel CD4-tdGFP and myr-RFP imaging, excitation was done at 800 nm.

Our chamber can be imaged in conjunction with both water and glycerol objectives with both upright and inverted microscopes. We compared the images of R7 growth cones of a P + 27% brain acquired by above setup with those acquired by a conventional Leica TCS SP5 confocal microscope with a resonant scanner, using a 63X Glycerol objective (NA = 1.3). A resonant scanner provides superior scan speeds compared to standard two-photon systems. However, we observed a decrease in signal intensity and quality with the confocal microscope at tissue deeper than 60 μm from the coverslip (*Figure 2—figure supplement 2i*). Even at moderately deep tissue, the laser power required on the confocal system to acquire images with comparable quality to the two-photon system is too high to take advantage of the higher scan speeds for extended periods (data not shown). Nevertheless, the resonant scanner would still be the preferred option for imaging at low depths when speed is the most important factor.

## Data analysis

Imaging data were analyzed and processed with Imaris (Bitplane). Deconvolved data were used in *Figures 4*, *5* and *6* and supplementary Videos. 3D deconvolution was performed with Autoquant X3 using adaptive PSF (blind) (*Hiesinger et al., 2001*). For all datasets, 10 iterations were performed at medium noise level (noise value: 20) with recommended settings. Distance from the coverslip was set to 40 μm.

Filopodial analysis was done with the Filament module of Imaris. Each filopodium was manually segmented and tracked across time points. 'Automatic placement' option was used while drawing to ensure that we measured the actual 3D length of each filopodium. We exported the 'length over time data for all of the filopodia of a growth cone to an Excel sheet and performed further analysis with MATLAB.

We used a custom MATLAB code to calculate the number of extension and retraction events, mean extraction and retraction speeds, mean lengths and lifetimes for each TrackID. Heat maps of lengths versus time for all filopodia in a growth cone were also generated. In those, filopodia were sorted by the angle of their orientation at the time of their initial formation. We did not find any overall, significant difference between the average speeds of extension and retraction on any growth cone; so they were combined to calculate a single average speed for all further analysis. We considered any changes in length less than 0.3 μm between consecutive time points as zero movement or 'static' periods because manual segmentation cannot be precise enough to reliably account for such a small retraction or extension. Average speeds were therefore calculated only from the points that had a change in length greater than 0.3 μm. We used the ratio of static time points to the lifetime of a filopodium to calculate 'inactivity' *Source code 1*.

Further analysis, i.e. classification into transient and stable filopodia, statistical analysis, and the generation of graphs were done with GraphPad Prism. Where needed statistical differences were calculated with unpaired, parametric t-tests. Filopodia number percentages over time in *Figure 4c* were fitted with second order polynomials to generate curves. For inactivity measurements, we generated two different graphs. Due to their short lifetimes, many transient filopodia have zero inactivity by definition; resulting in drastically lower average inactivity for transient filopodia compared to other filopodia (*Figure 4—figure supplement 2g–i*). This may unfairly imply an intrinsic difference of dynamics between transient and stable filopodia (*Figure 4—figure supplement 2j*). Indeed, when the filopodia with 'zero inactivity' are excluded, their average inactivity is statistically identical with the early-stage stable filopodia (*Figure 4—figure supplement 2k*). We therefore used these graphs in *Figures 4* and *6*.

We provide exemplary datasets in Zenodo (https://zenodo.org/record/33141). One file is a Zeiss (.lsm) file (raw data) and the other is the deconvolved version of this dataset in the Imaris (.ims) format, including the segmented filopodia as filament objects. This dataset belongs to the first fast

imaging session of P+28% growth cones shown in *Video 3*. We would be happy to provide any other datasets upon request.

## Acknowledgements

We would like to thank all members of the Hiesinger lab, Claude Desplan, Chi-Hon Lee, Iris Salecker, Steven Altschuler and Lani Wu for critical reading of the manuscript and for helpful discussions. We further thank Chi-Hon Lee, Lawrence Zipursky and the Bloomington Stock Center in Indiana for reagents. This work was supported by a grant from the National Institute of Health to PRH (RO1EY018884), the NeuroCure Cluster of Excellence and the FU Berlin (to PRH) as well as: VIB, University of Leuven, FWO, belspo, EMBO, and the European Commission's Marie Skłodowska Curie programs (to BAH).

## Additional information

### Funding

| Funder | Grant reference number | Author |
|---|---|---|
| National Eye Institute | EY018884 | P Robin Hiesinger |
| Freie Universität Berlin | | P Robin Hiesinger |
| University of Leuven | | Bassem A Hassan |
| Vlaams Instituut voor Biotechnologie | | Bassem A Hassan |
| EMBO | | Bassem A Hassan |
| Marie Curie program | | Bassem A Hassan |

The funders had no role in study design, data collection and interpretation, or the decision to submit the work for publication.

### Author contributions

MNÖ, Conception and design, Acquisition of data, Analysis and interpretation of data, Drafting or revising the article; ML, Analysis and interpretation of data; BAH, Conception and design, Drafting or revising the article; PRH, Conception and design, Analysis and interpretation of data, Drafting or revising the article

## Additional files

### Supplementary files

• Source code 1. The MATLAB function for analysis of the filopodial dynamics. This function takes as input the Excel files including the length and orientation data for all filopodia segmented across 60 time points within a 1 hour period. "TrackID"s are also required to identify the same filopodia across different time points. The function then calculates for each TrackID the number of extension and retraction events it experienced during that filopodium's lifetime, as well as the mean and standard deviation for extension, retraction and combined speeds. The user is asked for an input (in μm) defining the amount of extension/retraction length (default=0.3 μm) that will be considered insignificant, i.e. the filopodium will be assumed static for that 1 minute step. The function then outputs the number of extension and retraction events above that threshold ("filtered"), as well as speeds calculated from only these (above-threshold) events. The function also calculates the mean and standard deviation of a filopodium's length (μm) during its lifetime. These parameters are written in a new Excel file. Finally, the function creates the heat-maps used (and explained) in *Figure 4*.

### Major datasets

The following datasets were generated:

| Author(s) | Year | Dataset title | Dataset URL | Database, license, and accessibility information |
|---|---|---|---|---|
| Mehmet Neset Özel, Marion Langen, Bassem A Hassan, P Robin Hiesinger | 2015 | Filopodial dynamics and growth cone stabilization in Drosophila visual circuit development | https://zenodo.org/record/33141 | Publicly available at Zenodo (https://zenodo.org) |

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
