## [Decision Letter]

Thank you for submitting your work entitled "Filopodial Dynamics and Growth Cone Stabilization in *Drosophila* Visual Circuit Development" for peer review at *eLife*. Your submission has been favorably evaluated by K VijayRaghavan (Senior Editor), Christine Holt (Reviewing Editor), and three reviewers, one of whom, Carol Mason, has agreed to reveal her identity.

The reviewers have discussed the reviews with one another and the Reviewing editor has drafted this decision to help you prepare a revised submission.

The three reviewers find that the manuscript describes a very interesting set of data on the development and use of a new method of ex vivo culture and 2-photon confocal imaging for pupal CNS that enables a detailed analysis of neuronal development at subcellular resolution. It has long been known that insect head and compound eye development can proceed in culture and some acute imaging of the windowed or dissected brain has been performed previously (e.g. Medioni et al., 2015; Rabinovich et al., 2015) but the current study takes this simple system to a new and exciting level. The authors use this system in combination with MARCM clones to define the temporal periods and spatial properties of R-cell filopodial dynamics during target acquisition and stabilization in the optic lobes. Through careful comparison between fixed tissue and live imaging, they argue that many, if not most, features of R-cell connectivity are recapitulated in culture. This analysis of wild type genotypes reveals different phases of filopodial exploratory behavior in R cell growth cones. Moreover, analysis of *CadN* mutants also illustrates the value of live imaging, as some aspects of CadN function that have been speculated based on end-point or static analysis can be demonstrated. This is a very clear and well-documented methods paper with significant potential for future applications and for use of molecular markers and antibodies to define cell biological mechanisms underlying these dynamic processes. The reviewers agree that the work is of high quality (the movies are remarkable) and appropriate for publication in *eLife*. However, there are a few issues that authors should address, most of which are textual.

Essential revisions:

1) A major question raised by the reviewers was whether the filopodia described as 'growth cone filopodia' are, indeed, growth cone filopodia or whether they are axonal filopodia that emerge from the axon shaft during the phase of synapse formation during target recognition. The authors showed that *CadN* mutant R7 axons lack fast filopodial dynamics to conclude that fast filopodia dynamics are important for stabilization of axons. The conclusion seems to be well supported by the data presented. However, it is not clear whether the authors' claim that they have uncovered an unexpected role of filopodial dynamics in growth cone behavior is completely correct. Growth cone filopodia are thought to mediate axonal growth and guidance. By contrast, dendritic filopodia and axonal (those emanating from the axon shaft, not the terminal growth cone) filopodia seem to play a role in synaptogenesis and branch formation (e.g. work by Gallo and colleagues). A key question to be addressed here then is – are the axonal compartments analyzed in this paper during the target recognition phase of growth truly growth cones? This is of particular concern since the authors precisely showed that at the analyzed stages, R7 axonal terminals do not actively extend. Therefore, these axonal terminals might rather represent the transition from growth cone to an axonal compartment with filopodia involved in the establishment of adhesive-contacts (via CadN) between R7 and its target cells, which eventually will become nascent synapses.

2) On a related point, the authors do not discuss the possibility that rapid, transient filopodial dynamics reflect a signaling state that is required for synapse stabilization. Could this process provide or reflect the general signaling conditions needed for stability, or does this simply reflect a persistent state of plasticity/receptivity? The authors cannot conclude that the filopodia that they observe are functioning in the stabilization, but can claim a correlation with temporal periods and genotypes.

3) The last sentence of the Abstract is confusing: "Growth cone dynamics thereby underlie defects previously ascribed to incorrect target recognition". Do the authors mean that the defects in target recognition previously ascribed to mutations in molecules such as *CadN* or defects in entry/stabilization in the correct target zone can now be explained by defects in the dynamic behaviors of growth cones? Please clarify.

---

## [Author Response]

Essential revisions:

*1) A major question raised by the reviewers was whether the filopodia described as 'growth cone filopodia' are, indeed, growth cone filopodia or whether they are axonal filopodia that emerge from the axon shaft during the phase of synapse formation during target recognition. The authors showed that* CadN *mutant R7 axons lack fast filopodial dynamics to conclude that fast filopodia dynamics are important for stabilization of axons. The conclusion seems to be well supported by the data presented. However, it is not clear whether the authors' claim that they have uncovered an unexpected role of filopodial dynamics in growth cone behavior is completely correct. Growth cone filopodia are thought to mediate axonal growth and guidance. By contrast, dendritic filopodia and axonal (those emanating from the axon shaft, not the terminal growth cone) filopodia seem to play a role in synaptogenesis and branch formation (e.g. work by Gallo and colleagues). A key question to be addressed here then is – are the axonal compartments analyzed in this paper during the target recognition phase of growth truly growth cones? This is of particular concern since the authors precisely showed that at the analyzed stages, R7 axonal terminals do not actively extend. Therefore, these axonal terminals might rather represent the transition from growth cone to an axonal compartment with filopodia involved in the establishment of adhesive-contacts (via CadN) between R7 and its target cells, which eventually will become nascent synapses.*

We completely agree with the reviewers’ observation that our data show the transition of a growth cone to a more stable structure. To clarify our findings on filopodial dynamics during this transition, we now provide an extended analysis of the nature of filopodia observed in our study in comparison to axonal filopodia known from other systems in the Results section 3, including new Figure 3—figure supplement 1, as well as new corresponding Discussion sections, as outlined below.

A key advance and core idea of our study is the long-term imaging throughout the developmental phases from layer-specific targeting to synapse formation. As new layers form, the R7 growth cone undergoes shape changes that include a more typical growth cone morphology early on (see P+20-P+30% in new Figure 3—figure supplement 1), a bipolar structure with an added density in a newly forming layer (P+50%; red arrow in new Figure 3—figure supplement 1), thickening of the axon shaft and formation of extensions in layers where synapse formation occurs (P+60%-80% in new Figure 3—figure supplement 1). Before P+40%, we only observe filopodia at the axon tips as in classical growth cones. Therefore, we suggest these structures are still growth cones although they appear to use filopodia as a means to stabilize rather than as a substrate for migration. We think that the subsequent transition morphologies could reasonably be interpreted as an extended growth cone or a distinct and short part of the proximal axon that got recruited to new active functions during layer formation (Figure 3—figure supplement 1).

As rightly suggested in the reviewers’ comment, we find different types of filopodia at the terminal ending and axon shaft, including transiently on the axon in the first medulla layer (red arrow at P+40% in new Figure 3—figure supplement 1) and later throughout the medulla neuropil, but not on the axon leading to the medulla (P+60-80%, see Figure 3—figure supplement 1). We find these late filopodia more comparable to the axonal filopodia described by the Gallo group (e.g. Spillane et al., 2011 and 2012). However, the axonal filopodia observed in their ex vivo culture of the spinal cord appear to us more regularly spaced along long stretches of axon. In contrast, the filopodia we observed are extending only from specific regions of the axon-growth cone structure within the medulla neuropil, and never on the main axon leading to the neuropil. These filopodia are characteristic for the corresponding developmental processes in these layers, both spatially and temporally, as described in detail throughout the revised manuscript. We now provide a more detailed description of growth cone and filopodial dynamics and the Spillane et al. citations in the Results section 3 and in a separate paragraph in the Discussion section entitled ‘The role of filopodia’.

Finally, our data are consistent with the idea that CadN-mediated filopodia adhesion with target and other medulla cells is required for development; however, adhesion does not appear to specify contacts of R7 selectively with its target cells that might become nascent synapses, as suggested in the reviewers’ comment. First, CadN is present on numerous non-target cells throughout layer formation (e.g. Nern et al. 2008) and the R7 *CadN* mutant phenotype does not exhibit a selective reduction of specific filopodial contacts that could become nascent synapses, but really only a general slowdown of *all* filopodia throughout the entire layer formation process. The idea that CadN exerts a ‘non guidance cue’ function independent of its actual target is described in more detail in a revised final section of the Discussion.

*2) On a related point, the authors do not discuss the possibility that rapid, transient filopodial dynamics reflect a signaling state that is required for synapse stabilization. Could this process provide or reflect the general signaling conditions needed for stability, or does this simply reflect a persistent state of plasticity/receptivity? The authors cannot conclude that the filopodia that they observe are functioning in the stabilization, but can claim a correlation with temporal periods and genotypes.*

The idea that the >90% majority of filopodia that we describe as random and transient extension might implement a certain type of ‘signaling state’ is intriguing. In fact, in a recent paper by our groups on EGF receptor localization and signaling, we have described just such a mechanism for stochastic filopodial dynamics in EGF signaling (Zschaetzsch et al., 2014, *eLife*). However, at this point we cannot distinguish between an active signaling role and a ‘persistent state of plasticity/receptivity’, partly because we have not yet localized specific signaling receptors into the dynamic map. We further think we can rule out that especially the early transient filopodial dynamics play a role in synapse formation as they occur long before layer formation is concluded and synapse formation starts.

We further fully agree with the reviewers that our evidence for the role of these transient, random filopodial dynamics in stabilization is indeed based on correlation. However, we think the temporal sequence and nature of the live imaging data and collective evidence in wild type and CadN mutants provide a solid basis for our interpretation of the data and for further investigation of the precise implementation of stabilization, incl. possible functions through random filopodial dynamics, e.g. signaling. We now clearly state that‘our imaging data do not reveal a causal relationship between the observed dynamic behaviors’ in the Discussion section ‘The role of filopodia’.

*3) The last sentence of the Abstract is confusing: "Growth cone dynamics thereby underlie defects previously ascribed to incorrect target recognition". Do the authors mean that the defects in target recognition previously ascribed to mutations in molecules such as* CadN *or defects in entry/stabilization in the correct target zone can now be explained by defects in the dynamic behaviors of growth cones? Please clarify.*

We apologize for the lack of clarity – but yes, this is exactly what we were trying to say. The *CadN* mutant phenotype was previously described as a ‘targeting defect’ based on mistargeted axon termini in fixed preparations. Only live imaging of the dynamics could reveal that the mutant termini did not have a primary targeting defect, but are in fact destabilized after reaching to the correct target. This observation changes the interpretation of the function of CadN: it is clearly not required for targeting per se, but stabilizing what would otherwise be aberrant dynamic movements of the growth cone. In the revised manuscript we have changed the last sentence in the Abstract to: *‘*Hence, growth cone dynamics can influence wiring specificity without a direct role in target recognition and implement simple rules during circuit assembly.’